# Cell Transitions Contribute to Glucocorticoid-Induced Bone Loss

**DOI:** 10.3390/cells12141810

**Published:** 2023-07-08

**Authors:** Xiaojing Qiao, Xiuju Wu, Yan Zhao, Yang Yang, Li Zhang, Xinjiang Cai, Jocelyn A. Ma, Jaden Ji, Karen Lyons, Kristina I. Boström, Yucheng Yao

**Affiliations:** 1Division of Cardiology, David Geffen School of Medicine, University of California, Los Angeles, CA 90095, USA; xqiao@mednet.ucla.edu (X.Q.); xiujuwu@mednet.ucla.edu (X.W.); yanzhao@mednet.ucla.edu (Y.Z.); yanyang@mednet.ucla.edu (Y.Y.); liz@mednet.ucla.edu (L.Z.); xinjiangcai@mednet.ucla.edu (X.C.); jocelynma@g.ucla.edu (J.A.M.); jj152111@g.ucla.edu (J.J.); 2Department of Molecular, Cell & Developmental Biology at UCLA, Los Angeles, CA 90095, USA; klyons@mednet.ucla.edu; 3The Molecular Biology Institute at UCLA, Los Angeles, CA 90095, USA

**Keywords:** glucocorticoid, bone loss, osteoblast, transition, endothelial-like cells

## Abstract

Glucocorticoid-induced bone loss is a toxic effect of long-term therapy with glucocorticoids resulting in a significant increase in the risk of fracture. Here, we find that glucocorticoids reciprocally convert osteoblast-lineage cells into endothelial-like cells. This is confirmed by lineage tracing showing the induction of endothelial markers in osteoblast-lineage cells following glucocorticoid treatment. Functional studies show that osteoblast-lineage cells isolated from glucocorticoid-treated mice lose their capacity for bone formation but simultaneously improve vascular repair. We find that the glucocorticoid receptor directly targets Foxc2 and Osterix, and the modulations of Foxc2 and Osterix drive the transition of osteoblast-lineage cells to endothelial-like cells. Together, the results suggest that glucocorticoids suppress osteogenic capacity and cause bone loss at least in part through previously unrecognized osteoblast–endothelial transitions.

## 1. Introduction

More than seventy million adults worldwide continuously receive glucocorticoids for years to treat different kinds of medical conditions, such as transplant rejection, graft-versus-host disease, cancer, allergies, inflammatory and autoimmune disorders, and cardiovascular diseases [1,2,3]. Osteoporosis followed by severe fractures is one of the most prevalent side effects of this long-term therapy [1,2,3]. This type of osteoporosis is characterized by rapid bone loss upon initial glucocorticoid treatment followed by constant suppression of osteogenesis; higher doses of glucocorticoids result in more severe bone loss and fractures [1,2]. It has been shown that the hallmarks of osteoblast activity are dramatically decreased in glucocorticoid-induced bone loss [1,2].

Osteoblasts are responsible for bone formation. Residing at the endosteum and periosteum, osteoblasts maintain the plasticity to build bone [4,5,6]. Deficiency or abnormal differentiation of osteoblasts directly reduce the source of ossification [7,8,9,10]. In long-term glucocorticoid therapy, the number and activity of osteoblasts are significantly altered, causing bone to become more porous and fragile and leading to osteoporosis [1,2]. However, it is unknown if glucocorticoids drive osteoblasts to ill-fated transitions and derail osteoblastic differentiation to reduce osteogenesis.

Vascular endothelial cells coordinate with the osteogenic process to create the unique vasculature in bone [11,12,13,14,15], where endothelial cells interact with bone cells and bone cells regulate the differentiation and proliferation of endothelial cells [12,13,14,15]. Furthermore, the cell-fate transitions of endothelial cells have been uncovered in some medical conditions, such as vascular calcification or fibrodysplasia ossificans progressiva, where endothelial cells lose their identity and transition into osteoblast-like cells to form ectopic bone [16,17,18]. In this study, we find that glucocorticoids reciprocally convert osteoblast-lineage cells into endothelial-like cells. We hypothesize that glucocorticoids exert their suppressive effects on osteogenic capacity and cause osteoporosis in part through osteoblast–endothelial transitions.

## 2. Methods

### 2.1. Animals

*Wild type* (C57BL/6J), *Col1α1^CreERT2^* (B6.Cg-Tg(Col1α1-cre/ERT2)1Crm/J), and *Rosa^tdTomato^* (B6;129S6-Gt(ROSA)26Sortm9(CAG-tdTomato)Hze/J) on C57BL/6J background were obtained from the Jackson Laboratory. PCR was used to confirm genotypes [19], and generations F4–F6 were used to perform experiments. Littermates were used as experimental controls. A standard chow diet (Diet 8604, HarlanTeklad Laboratory, Indianapolis, Indiana, USA) was used to feed all mice. At 8 weeks of age, the mice received dexamethasone (0.08 µg/g, daily) for 4 weeks as previously described [20,21]. The Institutional Review Board reviewed and approved the studies, which were conducted in accordance with the animal care guidelines set by the University of California, Los Angeles (UCLA). The studies conformed to the National Research Council, *Guide for the Care and Use of Laboratory Animals, Eighth Edition* (Washington, DC, USA: The National Academies Press, 2011).

### 2.2. Tissue Culture

The osteoblast cell lines MC3T3 and hFOB1.19 were purchased from American Type Culture Collection (ATCC, CRL-2593 and CRL11372) and cultured as per the manufacturer’s protocol. Dexamethasone (Sigma-Aldrich, D1756. Saint Louis, MO, USA), Betamethasone (Sigma-Aldrich, B7005. Saint Louis, MO, USA), Triamcinolone (Sigma-Aldrich, 1676000. Saint Louis, MO, USA), Hydrocortisone (Sigma-Aldrich, H0888. Saint Louis, MO, USA), and Beclomethasone (Sigma-Aldrich, B0385. Saint Louis, MO, USA) treatments were performed as described in the experiments. Lentiviral vectors containing CMV-Osterix or Foxc2 siRNA were all purchased from GeneCopeia^TM^ (Rockville, MD, USA) and applied to the cells as per the manufacturer’s protocols.

### 2.3. Micro-CT Imaging

Mice were euthanized using methods approved by the Institutional Review Board. The femurs were cut free at the acetabulum and separated from the tibias. The attached muscle was removed gently from the femurs. Then, the femurs were fixed in 4% (wt/vol) paraformaldehyde (PFA) for 24 h. After 24 h, the PFA was replaced with 50% alcohol and the femurs were ready for Micro-CT imaging, which was performed at the Crump Imaging Center at UCLA. A high-resolution, volumetric micro-CT scanner (μCT125) was used to scan all the samples. The image data were acquired as previously described [19] with 10 μm isotropic voxel resolution, 200 ms exposure time, 2000 views, and 5 frames per view. A MicroView 3-D volume viewer and AltaViewer™ Software were used to analyze the images.

### 2.4. Laser Doppler Perfusion Image

A microcirculation imaging system (Perimed) was used to perform laser Doppler perfusion imaging as published [19]. Briefly, the imaging was conducted under normal ambient room lighting. A 20 × 27 mm high-resolution model with a 1388 × 1038 pixel measurement camera and 752 × 580 pixel documentation camera was used for one image per second. Settings of 20 µm/pixel and 21 images per frame were used until the imaging stopped. PIMSoft software (Perimed, Järfälla, Sweden) was used to analyze the data.

### 2.5. Mouse Surgery

A heated pad with a continuous flow of isoflurane was used for all the surgeries. Ectopic bone formation was performed as previously described [19]. Briefly, 5 × 10^5^ cells mixed with 40 mg hydroxyapatite/tricalcium phosphate powder (SALVIN^TM,^ ORASTRUCT-0.5CC. Charlotte, North Carolina, USA) were incubated in a 1 mL syringe at 37 °C in 5% CO_2_ overnight, then were transplanted into the pouch that was made on the back of mouse. Micro-CT imaging and histology were used to examine the implants at 12 weeks after transplantation.

The model of hindlimb ischemia was generated as published [19]. Briefly, the femoral artery was exposed and tied with double knots and 5 × 10^5^ cells were transplanted into the surgical area. Then, the incision was closed. Blood flow was monitored by laser Doppler perfusion imaging. The vascularization was examined through histology and immunostaining at 2 weeks after transplantation.

### 2.6. RNA Analysis and RNA-Sequencing

Real-time PCR analysis was performed as previously described [19]. Glyceraldehyde 3-phosphate dehydrogenase (GAPDH) was used as a control gene [19]. Primers and probes for mouse Osterix, Osteocalcin, Osteopontin, Foxc2, Flk1, VE-cadherin, and CD31 were obtained from Applied Biosystems (Waltham, MA, USA) as part of Taqman^®^ Gene Expression Assays.

For RNA-sequencing, total RNA was isolated using RNeasy Mini Kits (Qiagen, Hilden, Germany). The RNA libraries were constructed using a KAPA Stranded mRNA-Seq Kit (Roche, KK8420) and the sequencing was performed on the Illumina NovaSeq 6000 S4 platform (Illumina, Santa Monica, CA, USA) at the Technology Center for Genomics & Bioinformatics at UCLA. Spliced Transcripts Alignment to Reference (STAR, version 2.7.10a) was used to align the reads and count the number of reads per gene. We used the R package DESeq2 (version 1.36.0) to perform normalization and differential gene expression analysis. A log2 fold change of 1 and false discovery rate (FDR) of 0.05 were used as cut-offs to identify differentially expressed genes. RNA-seq data were deposited in the Gene Expression Omnibus (GEO) database.

### 2.7. Fluorescence-Activated Cell Sorting (FACS)

FACS analysis was performed as previously described [19]. Briefly, cleaned femurs were washed with PBS. The marrow was carefully flushed out with PBS using a syringe. Bones were cut into 1 to 2 mm lengths and incubated in collagenase solution (collagenase I and collagenase II mixture) for 25 min at 37 °C. Then, the solution was collected. The bones were washed with PBS three times to collect more cells. The combination of collected collagenase solution and PBS was filtered using cell strainers (70 µm and 40 µm). The filtered cell solution was centrifuged at 300× *g*. The supernatant was discarded, and 2 mL red blood cell lysis buffer was added to the precipitated cells for 10 min. Cells were collected again through centrifugation at 300× *g* and washed with PBS three times. For ectopic bone formation and vascular repair of hindlimb ischemia, tdTomato+ cells were isolated. For gene expression analysis, cells were fixed with 0.01% formaldehyde for 10–15 min and subjected to permeabilization with 0.1% Triton for 10 min. After that, cells were collected and new PBS was added, and cells were ready for incubation with the antibodies. The cells were stained with fluorescein isothiocyanate (FITC)-, phycoerythrin (PE)-, or Alexa Fluor 488 (AF-488)-conjugated antibodies against CD31 and VE-cadherin (all from BD biosciences, 550274 and 562243. Franklin Lakes, New Jersey, USA.) and Osteocalcin (ThermoFisher Scientific, 23418. Waltham, MA, USA). Nonspecific fluorochrome- and isotype-matched IgGs (BD Pharmingen, Franklin Lakes, New Jersey, USA.)) served as controls. A gating strategy was utilized as we used before [19]. Gates and regions were set up around the cell populations with forward scatter (FSC) and side scatter (SSC). Then, expressing markers were used to analyze these populations.

### 2.8. Immunoblotting and Immunofluorescence

Immunoblotting was performed as previously described [16]. Equal amounts of tissue lysates were used for immunoblotting. Blots were incubated with specific antibodies to Foxc2, CD31 (Abcam, ab5060, ab182982. Cambridge, UK), Osterix (Santa Cruz Biotechnology, sc-22536), and Osteocalcin (ThermoFisher Scientific, PA1-511A). β-actin (Sigma-Aldrich, A2228) was used as a loading control.

### 2.9. Immunofluorescence

Immunofluorescence was performed as previously described in detail [22]. Briefly, dissected bones were cleaned by removing muscle and fixed with 4% (wt/vol) ice-cold paraformaldehyde (PFA) solution for 4 h. The bones were washed with PBS three times to remove PFA and ice-cold 0.5 M EDTA, pH 7.4–7.6, was added for 24 h for decalcification. The bones were washed with PBS then incubated in cryoprotectant (CPT) solution for 24 h at 4 °C. After that, the bones were embedded and cryosectioned. We used specific antibodies to CD31 (BD Bioscience, 553370. Franklin Lakes, NJ, USA), Osterix (Santa Cruz Biotechnology, sc-22536), and vWF (Dako, A0082). The nuclei were stained with 4′,6-diamidino-2-phenylindole (DAPI, Sigma-Aldrich, D9564).

### 2.10. H&E Staining

Decalcified sections were prepared as described above for immunostaining [22]. Sections were washed with distilled water and nuclei were stained with the alum hematoxylin for 10 min. The sections were rinsed in running tap water and differentiated with 0.3% acid alcohol. After washing in tap water, the sections were stained with eosin for 2 min. After that, the sections were dehydrated, cleared, and mounted as previous described [22].

### 2.11. Statistical Analysis

GraphPad Instat^®^, version 3.0 (GraphPad Software), was used for statistical analysis. Either unpaired 2-tailed Student’s *t* test or one-way ANOVA with Tukey’s multiple-comparisons test were used to analyze for statistical significance.

## 3. Results

### 3.1. Endothelial Markers Emerge in the Osteoblast-Lineage Cells of Glucocorticoid-Induced Bone Loss

To determine if glucocorticoids drove osteoblast-lineage cells to unwanted cell differentiation, we generated a mouse model of glucocorticoid-induced osteoporosis by treating mice with dexamethasone (0.08 µg/g, daily) for 4 weeks as previously described [20,21]. Micro-computed tomography (micro-CT) imaging and histology confirmed the development of osteoporosis after the treatment (Figure 1a,b). We removed the marrow and isolated cells from the bone tissues. Fluorescence-activated cell sorting (FACS) revealed that a large cell population co-expressing Osteocalcin and endothelial markers CD31 or VE-cadherin appeared in the cells isolated from demarrowed bones of dexamethasone-treated mice (Figure 1c). Immunostaining showed that increased CD31 co-localized with Osteocalcin in the endosteum of the marrow cavity and trabecular bone surface, where osteoblasts reside (Figure 2a). High-magnification images of immunostaining also showed co-localization of Osteocalcin or Osterix with the endothelial markers VE-cadherin and von Willebrand factor (vWF) in the same locations (Figure 2b). We performed time-course experiments using FACS after 2 and 4 weeks of dexamethasone treatment. The results showed that the cell populations co-expressing Osteocalcin and endothelial markers CD31, VE-cadherin, or Flk1 appeared after 2 weeks of treatment and robustly increased after 4 weeks of treatment (Figure 2c). We further isolated the Osteocalcin-positive cells from the demarrowed bones and performed RNA sequencing. The result uncovered the up-regulation of the genes enriched in vascular endothelial differentiation, vasculature development, and angiogenesis in the dexamethasone-treated group, whereas the markers for ossification, bone development, and osteoblast differentiation were down-regulated (Figure 2d). Real-time PCR validated a significant decrease in osteoblast-lineage markers in Osteocalcin-positive cells of dexamethasone-treated mice compared to non-treated controls (Figure 2e). Together, the results suggested that an osteoblastic–endothelial transition occurred in glucocorticoid-induced bone loss.

We used an osteoblastic lineage tracing approach to further examine the osteoblast differentiation in glucocorticoid-induced bone loss. We used *Col1a1^cre/ERT2^Rosa^tdTomato^* mice in order to label the founder osteoblasts in vivo. The *Col1a1^cre/ERT2^* mice contained an inducible Cre driven by a 2.3 kb mouse *Col1a1* proximal promoter, which is active in all osteoblasts [23,24,25,26]. At 6 weeks of age, *Col1a1^cre/ERT2^Rosa^tdTomato^* mice received tamoxifen injections (75 µg/g, daily) for 5 consecutive days to induce tdTomato expression. At 8 weeks of age, tamoxifen-treated *Col1a1^cre/ERT2^Rosa^tdTomato^* mice received subcutaneous injections of dexamethasone (0.08 µg/g, daily) for 4 weeks to induce osteoporosis (Figure 3a–c). Saline-treated mice were used as controls. We examined the femurs of the dexamethasone-treated mice. Immunostaining revealed that the increased VE-cadherin was expressed in the tdTomato-positive cells, which were derived from the tdTomato-labeled founder osteoblasts (Figure 3d). FACS analysis also identified a group of tdTomato-positive cells that expressed VE-cadherin or CD31 in the cell population isolated from the femurs (Figure 3e). The results suggested that glucocorticoids drove osteoblast-lineage cells toward endothelial differentiation in glucocorticoid-induced bone loss.

*Col1a1^cre/ERT2^* mice contain Col1a1 promoter-driven CreER, where Cre is fused to a modified estrogen receptor (ER) [27]. In these mice, CreER is expressed in the cells where Col1a1 is expressed. However, without tamoxifen, CreER remains in the cytoplasm without entering the nucleus to accomplish the recombination of target genes [27]. Besides bone, Col1a1 is expressed in other tissues where CreER expression could be activated in the *Col1a1^cre/ERT2^* mice [23]. We systemically examined the CreER expression in different organs, including bone, lungs, skin, heart, muscle, bone marrow, liver, kidneys, stomach, spleen, aorta, eyes, small intestine, pancreas, and brain, of *Col1a1^cre/ERT2^* mice, *Col1a1^cre/ERT2^Rosa^tdTomato^* mice, *Col1a1^cre/ERT2^Rosa^tdTomato^* mice with tamoxifen treatment, and *Col1a1^cre/ERT2^Rosa^tdTomato^* mice with tamoxifen and dexamethasone treatment. Real-time PCR showed the highest expression of CreER in bone with only low levels of CreER expression in other tissues (Figure 4a), which confirmed a previous study that showed osteoblasts to have the highest CreER expression [23]. The results also revealed that the treatment with tamoxifen or dexamethasone had no effect on CreER expression in all these organs (Figure 4a).

In *Col1a1^cre/ERT2^Rosa^tdTomato^* mice, the administration of tamoxifen translocated CreER into the nuclei of the cells that expressed Col1a1 and activated tdTomato expression to label these cells. We found that glucocorticoids drove the tdTomato-positive osteoblast-lineage cells toward endothelial differentiation (Figure 3). To determine if tdTomato-positive marrow cells in bone or tdTomato-positive cells of other organs were involved in the osteoblast–endothelial transition, we first used FACS to examine the percentage of tdTomato-positive cells in the bone, lungs, skin, heart, muscle, bone marrow, liver, kidneys, stomach, spleen, aorta, eyes, small intestine, pancreas, and brain of *Col1a1^cre/ERT2^Rosa^tdTomato^* mice, *Col1a1^cre/ERT2^Rosa^tdTomato^* mice with only tamoxifen treatment, and *Col1a1^cre/ERT2^Rosa^tdTomato^* mice with tamoxifen and dexamethasone treatment. The results showed the highest percentage of tdTomato-positive cells in bone, with only low percentage of tdTomato-positive cells in other tissues of *Col1a1^cre/ERT2^Rosa^tdTomato^* mice with only tamoxifen treatment and *Col1a1^cre/ERT2^Rosa^tdTomato^* mice with tamoxifen and dexamethasone treatment (Figure 4b). The distribution pattern was similar to that of the CreER expression (Figure 4b), suggesting that tamoxifen efficiently moved CreER into cell nuclei for tdTomato activation. The results did not show a significant difference in percent of tdTomato-positive cells between dexamethasone treatment and the control, suggesting that glucocorticoids had no effect on CreER translocation or tdTomato activation (Figure 4b). We then isolated tdTomato-positive cells from bone, lungs, skin, heart, muscle, bone marrow, liver, kidneys, stomach, spleen, and aorta, and examined the endothelial markers VE-cadherin, CD31, Flk1, and Foxc2. We only found induction of these markers in tdTomato-positive cells from bone, not in cells from bone marrow or other tissues (Figure 4c). The results suggested that glucocorticoid-driven endothelial differentiation only occurred in tdTomato-labelled osteoblast-lineage cells and therefore ruled out the involvement of marrow progenitors or mesenchymal cells from other organs.

To further determine if glucocorticoids drove the osteoblast–endothelial transition causing bone loss, we performed a second osteoblastic lineage tracing using Osterix promotor-driven GFP transgenic (*OSX-Gfptg*) mice [28]. We treated the *OSX-Gfptg* mice with dexamethasone (0.08 µg/g, daily) for 4 weeks and validated the osteoporosis (Figure 5a,b). Immunostaining and FACS analysis showed increased CD31 expression in the GFP-positive cells, which were derived from GFP-labeled founder osteoblast-lineage cells (Figure 5c,d). Together, the results supported that glucocorticoids drove osteoblast-lineage cells toward endothelial differentiation, leading to bone loss.

### 3.2. Glucocorticoid-Treated Osteoblasts Lose Osteogenic Capacity in Ectopic Bone Formation but Improve Vascular Repair

To determine changes in the osteoblast capacity in glucocorticoid-induced osteoporosis, we performed transplantation experiments and examined the capacity using well-established assays in vivo (Figure 6a, top). We isolated tdTomato-positive cells from the femurs of *Col1a1^cre/ERT2^Rosa^tdTomato^* mice, which were prepared with the same treatment of tamoxifen and dexamethasone for osteoblast-lineage labelling and osteoporosis induction (Figure 6a, bottom). Since the tdTomato expression was induced prior to the dexamethasone treatments, the results showed similar tdTomato levels between the treated group and non-treated control, suggesting similar efficiency of tdTomato labelling in osteoblast-lineage cells in these groups. We examined the osteogenic markers Osterix, Osteopontin, and Osteocalcin in tdTomato-positive cells, and found a remarkable decrease in these markers in the dexamethasone-treated group (Figure 6b). The cells were then transplanted into nude mice to examine bone formation. Three weeks after transplantation, micro-CT imaging showed a smaller size of ectopic bone formation with less bone volume in the implants of the dexamethasone-treated group than the controls (Figure 6c). Immunostaining showed more CD31+ cells in the implants of the dexamethasone-treated group than the controls (Figure 6d). The results suggested a reduction in osteogenic capacity with induction of endothelial characteristics in the osteoblast-lineage cells of glucocorticoid-induced bone loss.

We again isolated tdTomato-positive cells from the femurs of *Col1a1^cre/ERT2^Rosa^tdTomato^* mice, which were pre-treated with tamoxifen and dexamethasone, and examined the endothelial markers Foxc2, Flk1, VE-cadherin, and CD31. We found an induction of these markers in the dexamethasone-treated group compared to controls (Figure 6e). After ligation of the proximal and distal femoral artery, we transplanted tdTomato-positive cells into a model of hindlimb ischemia in nude mice to evaluate the angiogenic capacity of the tdTomato-positive cells. Laser Doppler perfusion imaging demonstrated significantly higher limb blood flow in the mice transplanted with tdTomato-positive cells from the dexamethasone-treated group than controls (Figure 6f). Latex dye staining with tissue clearing showed stronger vascularization at the site of transplantation of the dexamethasone-treated group than that of the non-treated group (Figure 6g). The results suggested that the osteoblast-lineage cells in glucocorticoid-induced osteoporosis gained angiogenic capacity to improve vascular repair.

### 3.3. The Coupled Alterations in Foxc2 and Osterix Are Responsible for the Shift of Osteoblast-Lineage Cells to Endothelial Differentiation

To understand how glucocorticoids drove the osteoblast-lineage cells toward unwanted differentiation, we obtained a mouse osteoblast line (MC3T3-E1) that has been broadly used for osteoblast studies. We treated the osteoblasts with 10^−7^ M glucocorticoids, including dexamethasone, triamcinolone, betamethasone, beclomethasone, and hydrocortisone. After 24 h treatments, the FACS results showed that these glucocorticoids strongly induced the expression of the endothelial marker VE-cadherin in the osteoblasts (Figure 7a). We then examined the time-course expression of endothelial and osteogenic markers in the osteoblast-lineage cells after dexamethasone treatment. The results showed that an increase in Foxc2 and a decrease in Osterix occurred ahead of the changes in other markers (Figure 7b). We also obtained a human osteoblast line (hFOB1.19) and treated the cells with 10^−7^ M dexamethasone. Real-time PCR showed similar pattern of the time-course expression of endothelial and osteogenic markers after dexamethasone treatment (Figure 7c). The results suggested that the alterations in Osterix and Foxc2 were important for osteoblasts to lose their cell fate and undergo endothelial differentiation.

Foxc2 and Osterix are the early drivers for endothelial and osteogenic differentiation, respectively, and we have shown that Foxc2 was upregulated by glucocorticoids whereas Osterix was downregulated (Figure 2c and Figure 6b,d), suggesting important roles of Foxc2 and Osterix in the shift of osteoblasts in glucocorticoid-induced osteoporosis. To further determine if the glucocorticoid-altered expression of Foxc2 and Osterix was critical to drive osteoblasts to endothelial differentiation, we directly changed the levels of Foxc2 and Osterix in dexamethasone-treated osteoblasts using lentiviral vectors containing Foxc2 siRNA or cytomegalovirus promoter-driven (CMV) Osterix cDNA, respectively. The results showed that the knockdown of Foxc2 alone abolished the induction of endothelial markers but failed to prevent the reduction in osteoblastic markers (Figure 8a,b). Overexpression of Osterix prevented the reduction in osteoblast markers but was unable to abolish the endothelial markers (Figure 8a,b). Only the combination of Foxc2 knockdown and Osterix overexpression prevented dexamethasone from inducing endothelial markers and reducing osteoblastic markers, suggesting that the coupled modulations of Foxc2 and Osterix were required to shift the osteoblasts to endothelial differentiation.

## 4. Discussion

In this study, we find a shift of osteoblast-lineage cells toward endothelial differentiation induced by glucocorticoids and show that this unwanted cell transition causes osteoblast-lineage cells to lose osteogenic capacity, thereby contributing to glucocorticoid-induced bone loss. The results suggest that glucocorticoids suppress osteogenic capacity and cause osteoporosis in part through this previously unrecognized transition of osteoblast-lineage cells to endothelial differentiation. This study provides a new mechanism and insights into glucocorticoid-induced bone loss for counteracting this toxic effect of long-term glucocorticoid therapy.

Osteoblasts are derived from mesenchymal progenitors and differentiate into osteocytes to build bone. However, the manipulation of osteoblast plasicity has been reported in previous studies. For example, reciprocal transitions between osteoblasts and adipocytes have been demonstrated in multiple studies [29,30,31,32,33,34,35,36]. An inhibitor of glycogen synthase kinase 3 also alters the differentiation of osteoblastic-like cells to improve the vascular calcification [19]. Here, we labelled the osteoblast-lineage cells in vivo and found that glucocorticoids stimulated osteoblast-lineage cells towards endothelial differentiation. We also showed that coupled alterations in Foxc2 and Osterix are responsible for the transition of osteoblast-lineage cells to endothelial differentiation.

Foxc2 belongs to the family of forkhead-box transcription factors [37] and is required for the development of vasculature [38]. Foxc2 is observed in all endothelial lineages during development, such as endothelial progenitors, and arterial, venous, and lymphatic endothelial cells [39,40,41]. Acting as an early driver, Foxc2 governs the specification of arterial endothelial cells and controls angiogenesis and lymphatic vessel formation [37,38,39,40,41]. As expected, Foxc2-deficient mice exhibit severe defects in the lymphatic valves and vasculature [41]. Specific deletion of Foxc2 in endothelial cells causes malformation of microvessels [42]. In this study, we find that glucocorticoids induce Foxc2 in osteoblast-lineage cells. Coupled with alteration of Osterix, induced Foxc2 contributes to shifting the osteoblast-lineage cells towards endothelial transition.

Osterix, also named Sp7, is a zinc-finger transcription factor and well known to regulate bone formation [8]. Osterix drives mesenchymal precursors to differentiate into osteoblasts and osteocytes [43,44,45]. Mice with Osterix gene deletion show no development of bone tissue [8]. Abnormal expression of Osterix is also associated with bone-related medical conditions in humans, such as osteogenesis imperfecta, rheumatoid arthritis, bone fracture repair, and osteoporosis [46,47,48,49]. Here, we show that glucocorticoids suppress Osterix expression. Together with the activation of Foxc2, the suppression of Osterix leads to switching the cell fate and functional capacity in osteoblast-lineage cells, contributing to osteoporosis. These results may provide novel strategies for treating glucocorticoid-induced bone loss by maintaining the levels of Foxc2 and Osterix.

## Figures and Tables

**Figure 1 cells-12-01810-f001:**
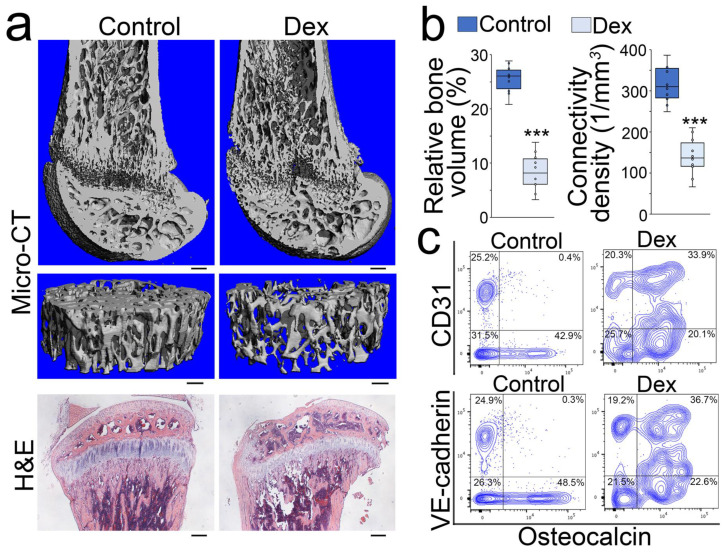
Endothelial markers emerge in the osteoblasts of glucocorticoid-induced bone loss. (**a**): Micro-CT and H&E staining of the mouse femurs after dexamethasone (Dex) treatment (n = 10). Control, saline. Scale bar, 200 µm. (**b**): Relative bone volume and connective density of mouse femurs after dexamethasone or control treatment (n = 10). Relative bone volume was calculated using bone volume/total volume based on counting voxels. The data were analyzed for statistical significance using unpaired 2-tailed Student’s t test. The bounds of the boxes represent the upper and lower quartiles with data points. The line in the box is the median. Error bars are maximal and minimal values. ***, *p* < 0.0001. (**c**): FACS analysis of demarrowed femurs of dexamethasone-treated mice using anti-CD31, anti-VE-cadherin, or anti-Osteocalcin antibodies.

**Figure 2 cells-12-01810-f002:**
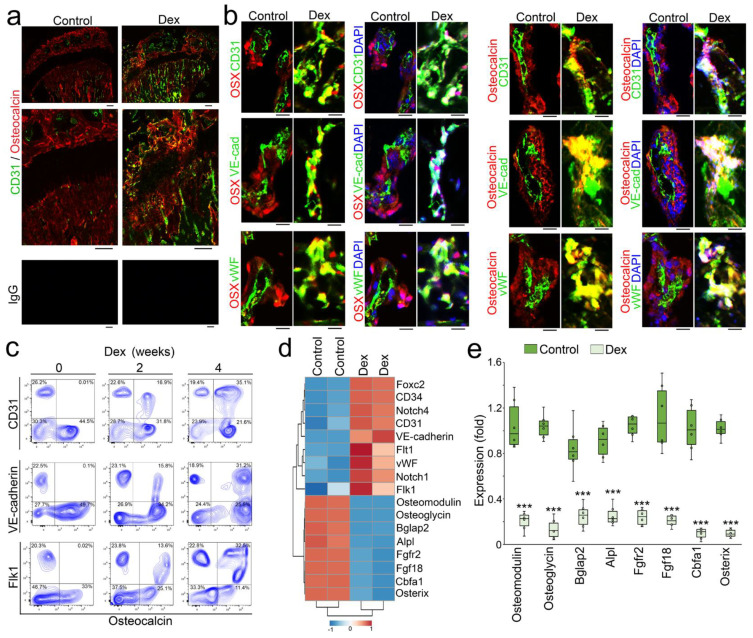
Endothelial markers emerge in the osteoblast-lineage cells of glucocorticoid-induced bone loss. (**a**): Immunostaining of the femurs of dexamethasone-treated mice using anti-CD31 (green) and anti-Osteocalcin antibodies. Scale bar, 100 µm. (**b**): High-magnification images of immunostaining of the femurs of dexamethasone-treated mice using anti-CD31, anti-VE-cadherin, or anti-vWF antibodies (green) with anti-Osterix (OSX) or anti-Osteocalcin antibodies (red). DAPI (blue) was used for nuclei staining. Scale bar, 20 µm. (**c**): Time-course FACS analysis of demarrowed femurs of dexamethasone-treated mice using anti-CD31, anti-VE-cadherin, anti-Flk1, or anti-Osteocalcin antibodies. (**d**): Heatmap of representative osteogenic and endothelial markers using normalized and log2-transformed gene counts. (**e**): Real-time PCR results of osteoblast-lineage markers (n = 6). The data were analyzed for statistical significance using unpaired 2-tailed Student’s t test. The bounds of the boxes represent the upper and lower quartiles with data points. The line in the box is the median. Error bars are maximal and minimal values. ***, *p* < 0.0001.

**Figure 3 cells-12-01810-f003:**
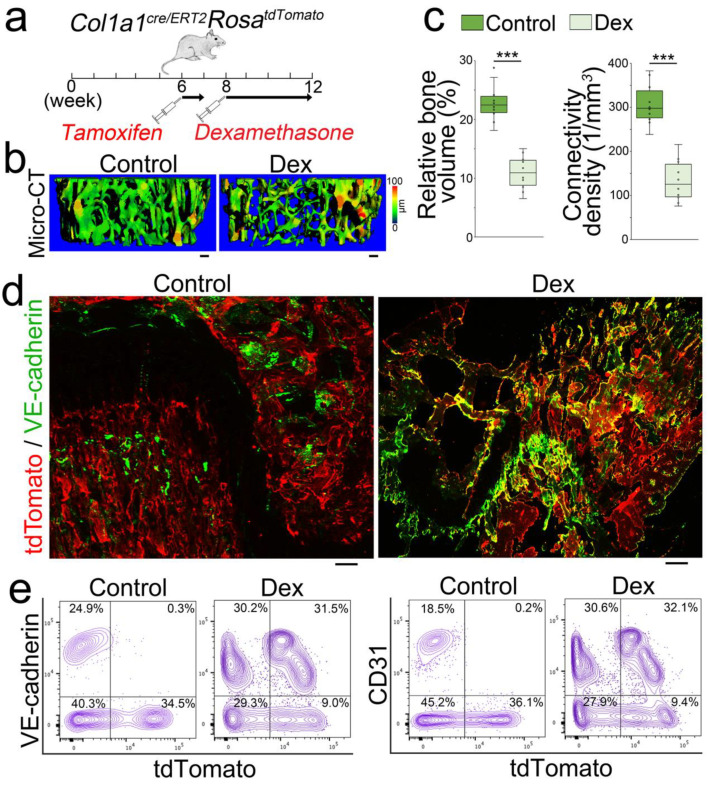
Osteoblastic lineage tracing identifies endothelial markers in the osteoblasts of glucocorticoid-induced osteoporosis. (**a**): Schematic of experimental procedure for labelling founder osteoblasts in glucocorticoid-induced osteoporosis. (**b**): Micro-CT of the femurs of Col1a1cre/ERT2RosatdTomato mice after dexamethasone (Dex) treatment. Control, saline. Scale bar, 100 µm. (**c**): Relative bone volume and connective density of mouse femurs after dexamethasone treatment (n = 10). Control, saline. The data were analyzed for statistical significance using unpaired 2-tailed Student’s t test. The bounds of the boxes represent upper and lower quartiles with data points. The line in the box is the median. Error bars are maximal and minimal values. ***, *p* < 0.0001. (**d**): The expression of tdTomato (red) and VE-cadherin (green) in the femurs of dexamethasone-treated mice. VE-cadherin was detected by immunostaining using anti-VE-cadherin (green). Scale bar, 100 µm. (**e**): Co-expression of tdTomato with endothelial markers detected through FACS analysis of demarrowed femurs of dexamethasone-treated mice using anti-CD31 and anti-VE-cadherin.

**Figure 4 cells-12-01810-f004:**
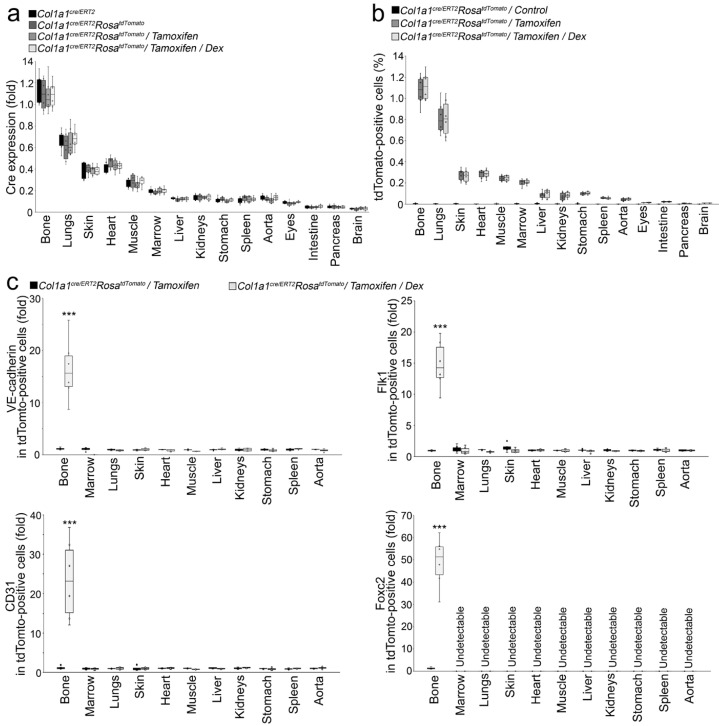
Dexamethasone had no effect on the CreER expression or tdTomato activation, and dexamethasone did not activate the expression of endothelial markers in other tissues. (**a**): Cre expression in different tissues (n = 6). (**b**): TdTomato-positive cells in different tissues (n = 6). (**c**): Expression of endothelial markers in tdTomato-positive cells isolated from different tissues (n = 6). Data were analyzed for statistical significance using ANOVA with post hoc Tukey’s analysis. The bounds of the boxes are upper and lower quartiles with data points. The line in the box is the median. Error bars are maximal and minimal values. ***, *p* < 0.0001.

**Figure 5 cells-12-01810-f005:**
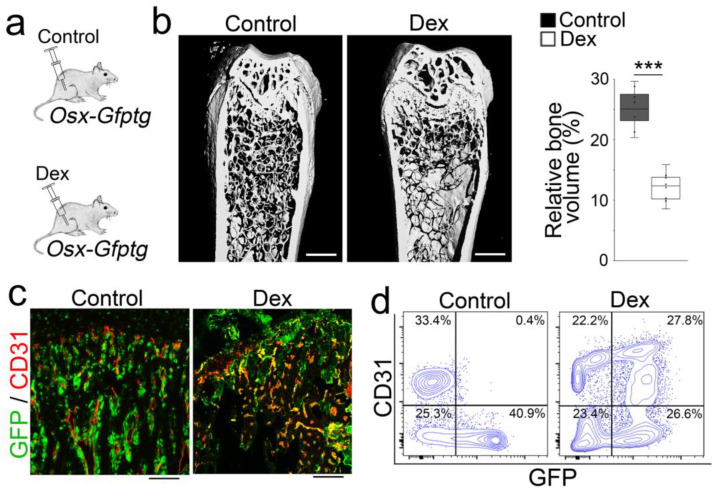
Identified endothelial markers in the osteoblasts of glucocorticoid-induced osteoporosis of OSX-Gfptg mice. (**a**): Schematic of experimental procedure for labelling founder osteoblasts in glucocorticoid-induced osteoporosis of *OSX-Gfptg* mice. (**b**): Micro-CT and relative bone volume of the femurs of *OSX-Gfptg* mice after dexamethasone (Dex) treatment. Control, saline. Scale bar, 100 µm. The data were analyzed for statistical significance using unpaired 2-tailed Student’s t test. The bounds of the boxes represent upper and lower quartiles with data points. The line in the box is the median. Error bars are maximal and minimal values. ***, *p* < 0.0001. (**c**): The expression of GFP (green) and CD31 (red) in the femurs of dexamethasone-treated mice. CD31 was detected by immunostaining using anti-CD31 (red). Scale bar, 100 µm. (**d**): Co-expression of GFP with endothelial markers detected through FACS analysis of demarrowed femurs of dexamethasone-treated mice using anti-CD31 antibodies.

**Figure 6 cells-12-01810-f006:**
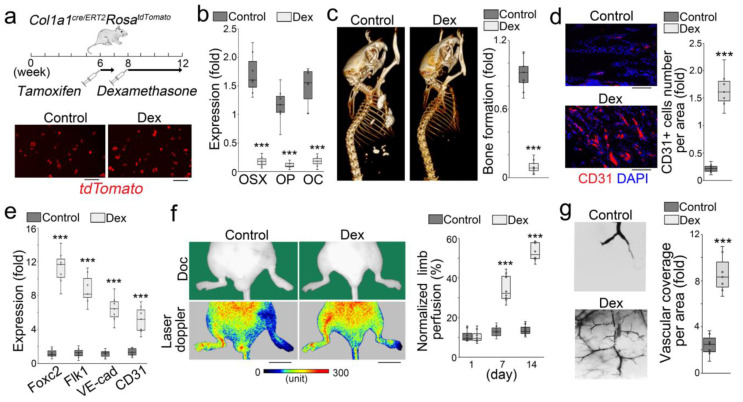
Osteoblasts of glucocorticoid-induced osteoporosis lose osteogenic capacity but improve vascular repair. (**a**): Schematic of experimental procedure (top) and tdTomato+ cells isolated from the femurs of *Col1a1^cre/ERT2^Rosa^tdTomato^* mice, which were prepared with osteoblast-lineage labelling and osteoporosis induction (bottom). Scale bar, 50 µm. Control, saline treatment. (**b**): Expression of osteogenic markers in tdTomato-labelled osteoblasts of the demarrowed femurs of dexamethasone-treated mice (n = 7). OSX, Osterix; OP, Osteopontin; OC, Osteocalcin. (**c**): Micro-CT images of ectopic bone formation with relative volume in nude mice after cell transplantation of tdTomato-labelled osteoblasts isolated from femurs of dexamethasone-treated mice. (**d**): Immunostaining using anti-CD31 antibodies with quantitation of CD31+ cells in the implants of tdTomato-labelled osteoblasts (n = 7). (**e**): Expression of endothelial markers in tdTomato-labelled osteoblasts of the demarrowed femurs of dexamethasone-treated mice (n = 7). VE-cad, VE-cadherin. (**f**): Laser Doppler perfusion images and percentage of blood flow perfusion in hindlimb ischemia model after transplantation of tdTomato-labelled osteoblasts isolated from the femurs of dexamethasone-treated mice. Top, documentation camera (doc). Bottom, measurement camera (laser Doppler). Scale bar, 10 mm. (**g**): Latex dye staining with tissue clearing and analysis of vascular coverage at the sites of transplantation of tdTomato-labelled cells (n = 7). (**b**–**e**,**g**) were analyzed for statistical significance using unpaired 2-tailed Student’s t test. f was analyzed for statistical significance using ANOVA with post hoc Tukey’s analysis. The bounds of the boxes are upper and lower quartiles with data points. The line in the box is the median. Error bars are maximal and minimal values. ***, *p* < 0.0001.

**Figure 7 cells-12-01810-f007:**
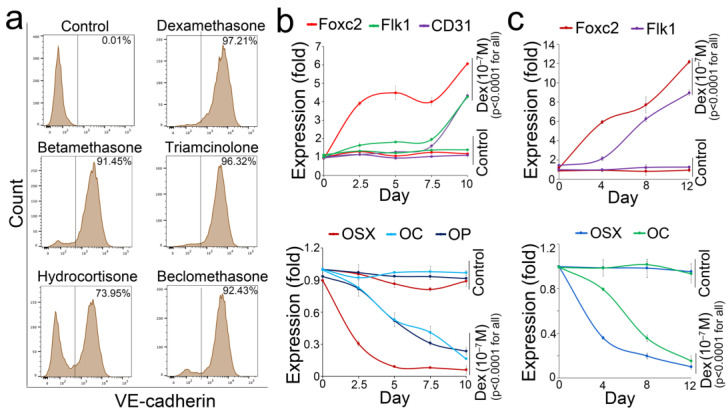
Coupled alterations in Foxc2 and Osterix are responsible for glucocorticoid shifting osteoblasts toward endothelial differentiation. (**a**): FACS of the expression of VE-cadherin in osteoblasts after different glucocorticoid treatments. (**b**): Time-course expression of endothelial and osteogenic markers in dexamethasone-treated mouse osteoblasts (n = 8). (**c**): Time-course expression of endothelial and osteogenic markers in dexamethasone-treated human osteoblasts (n = 7). The data were analyzed for statistical significance using ANOVA with post hoc Tukey’s analysis. Error bars are maximal and minimal values.

**Figure 8 cells-12-01810-f008:**
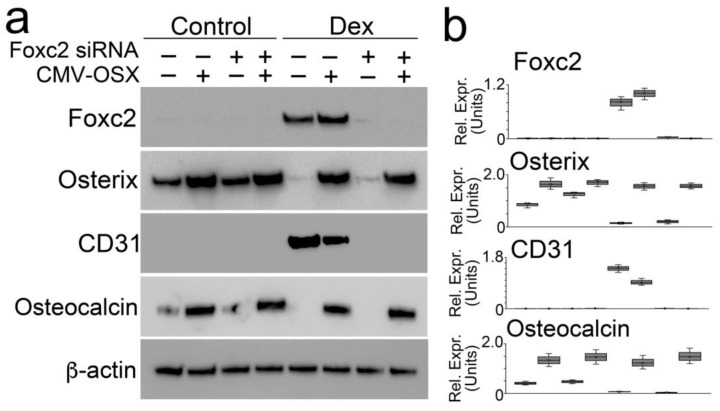
Coupled alterations in Foxc2 and Osterix are responsible for glucocorticoid shifting osteoblasts toward endothelial differentiation. (**a**): Immunoblotting with densitometry analysis of Foxc2, Osterix, Osteocalcin, and CD31 in dexamethasone-treated osteoblasts after Foxc2 knockdown (Foxc2 siRNA) or Osterix overexpression (CMV-OSX). β-actin was used as loading control. (**b**): Densitometry analysis of immunoblotting.

## Data Availability

RNA-sequencing data were deposited in the GEO database with access number GSE216891.

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
