# Peer review of "Cell Transitions Contribute to Glucocorticoid-Induced Bone Loss"

_cells, 2023, doi:10.3390/cells12141810_

Round 1

Reviewer 1 Report

Qiao et al. described that a possible aspect of cell fate transition from osteoblasts to endothelial-like cells in Glucocorticoid-induced bone loss model. This is a very interesting and novel approach to show that osteoblasts can change their cell fate towards endothelial cells upon glucocorticoid treatment. However, there are several concerns that have to be addressed.

Authors showed osteoblasts-endothelial cell transition based on few marker expression such as CD31 and VE-cadherin. Due to the adherent cell character of the osteoblasts, it is known that osteoblasts express several cadherins. Some more markers for endothelial cells and/or pericytes can be helpful to define the cell type.  

Also, I wonder if Dex treated mice have more abundant endothelial cells than osteoblasts, whether they have more vasculature compared to the control mice (e.g. Type H vessels formation).

Fig2. Can the authors indicate the locations of the magnified pictures ( b) in the picture (a)?  Are we seeing on the similar region of interest? And osteocalcin staining seems to have a high background. Can the authors show the figure with nuclei staining, so it will be possible to distinguish single cells?

Line 183: “osteocalcin in the endosteum of the marrow cavity.” This reviewer can see the osteocalcin labeling not only on the endosteum, but also mature osteoblasts that are residing on the trabecular bone surface. Please rewrite it clearly.

Fig2c. If the cells are sorted based on osteocalcin expressing cells, why Bglap2 and other OB markers expressed at a low level in the Dex groups? Can the authors show any validation experiments at least by qPCR?

Fig3a. Experimental scheme for treatments and your text in the manuscript (Line 215-218) are not matching.

Fig6a: tdTomato labeled images -> cells in vitro? Figure legend and main text should be mentioned clearly.

Fig6: If the osteoblast lineage cells (tdTomato) lose their osteogenic capacity, but enhance endothelial cell capacity, can the authors show that the ectopic bone formation using Dex-treated tdTomato cells shows more enriched vascularization instead of bone? Also, I wonder if the tdTomato labeled cells are involved in angiogenesis to repair the vasculature in the ischemic mouse system (Fig 6e). Can the authors show microscopic images of the healed vascular?

Author Response

Reviewer 1 

Qiao et al. described that a possible aspect of cell fate transition from osteoblasts to endothelial-like cells in Glucocorticoid-induced bone loss model. This is a very interesting and novel approach to show that osteoblasts can change their cell fate towards endothelial cells upon glucocorticoid treatment. However, there are several concerns that have to be addressed.

1.“Authors showed osteoblasts-endothelial cell transition based on few marker expression such as CD31 and VE-cadherin. Due to the adherent cell character of the osteoblasts, it is known that osteoblasts express several cadherins. Some more markers for endothelial cells and/or pericytes can be helpful to define the cell type.” 

In addition to the expression of CD31 and VE-cadherin, we showed the expression of vWF co-localized with Osterix and Osteocalcin (Figure 2b). We added time-course FACS analysis of demarrowed femurs of dexamethasone-treated mice using anti-Flk1, anti-CD31, anti-VE-cadherin with anti-Osteocalcin antibodies. (Figure 2c). In RNA-sequencing, we showed the induction of the endothelial markers Notch1 and 4, CD34, Foxc2, Flt1 in Osteocalcin-positive cells.

  1. “Also, I wonder if Dex treated mice have more abundant endothelial cells than osteoblasts, whether they have more vasculature compared to the control mice (e.g. Type H vessels formation).”

Dexamethasone treatment caused a complicated shift of cells in the bone. FACS results of whole demarrowed femur bone showed a decrease in cell populations expressing only Osteocalcin or only CD31 or only VE-cadherin, suggesting that a decrease of both mature osteoblasts and ECs after Dexamethasone treatment. The abundance of cells expressing only Osteocalcin was higher than that of cells expressing only CD31 or VE-cadherin, suggesting the abundance of normal osteoblasts was higher than that of normal ECs. Cell populations co-expressing Osteocalcin and CD31 or VE-cadherin increased after Dexamethasone treatment. However, these cells were derived from osteoblast lineage cells and may not count as normal ECs, and they may not form normal vessels in the bone.

  1. “Fig2. Can the authors indicate the locations of the magnified pictures ( b) in the picture (a)?  Are we seeing on the similar region of interest? And osteocalcin staining seems to have a high background. Can the authors show the figure with nuclei staining, so it will be possible to distinguish single cells?”

We performed immunostaining using a number of sections in each experimental group.  The images of low and high magnification were captured from different sections from same group. We have examined the images of osteocalcin staining and confirmed that the images were taken with same exposure time. Co-staining with DPAI for nuclei detection has been added to Figure 2b.

  1. “Line 183: “osteocalcin in the endosteum of the marrow cavity.” This reviewer can see the osteocalcin labeling not only on the endosteum, but also mature osteoblasts that are residing on the trabecular bone surface. Please rewrite it clearly.”

We have revised the sentence.

  1. “Fig2c. If the cells are sorted based on osteocalcin expressing cells, why Bglap2 and other OB markers expressed at a low level in the Dex groups? Can the authors show any validation experiments at least by qPCR?”

We performed real-time PCR to validate the alterations in osteoblast lineage markers. The result has been added to Figure 2e and showed a decrease in osteoblast lineage markers in the dexamethasone treated group compared to control.

  1. “Fig3a. Experimental scheme for treatments and your text in the manuscript (Line 215-218) are not matching.”

Thank you for catching this. We have revised Figure 3a.

  1. “Fig6a: tdTomato labeled images -> cells in vitro? Figure legend and main text should be mentioned clearly.”

The legend of Figure 6a and main text have been revised.

  1. “Fig6: If the osteoblast lineage cells (tdTomato) lose their osteogenic capacity, but enhance endothelial cell capacity, can the authors show that the ectopic bone formation using Dex-treated tdTomato cells shows more enriched vascularization instead of bone? Also, I wonder if the tdTomato labeled cells are involved in angiogenesis to repair the vasculature in the ischemic mouse system (Fig 6e). Can the authors show microscopic images of the healed vascular?”

We performed immunostaining using anti-CD31 antibodies to quantitate CD31+ cells in the implants of tdTomato-labelled osteoblasts. The results showed more CD31+ cells in the implants of the dexamethasone-treated group that the non-treated control (Figure 6d). We also examined the vascularization at the sites of transplantation of tdTomato-labelled cells using latex dye with tissue clearing, which showed stronger vascular growth in the dexamethasone-treated group than that of the non-treated control (Figure 6g).

Reviewer 2 Report

The manuscript entitled ”Cell Transitions Contribute to Glucocorticoid-Induced Bone Loss” by Qiao et al. addresses one major unwanted side effect of glucocorticoid treatment: glucocorticoid-induced osteoporosis, which may increase bone fracture risk. The authors describe that glucocorticoids convert osteoblast-lineage cells (OBs) into endothelial-like cells (ECs).

The authors first identified the well-known osteoporotic effect of dexamethasone in C57BL/6J mice. They surprisingly identified endothelial markers VE-cadherin and CD31, as well as vWF, which emerged in the osteoblast lineage cells identified via osteocalcin and osterix.
To identify the origin of the EC markers from osteoblasts, the authors used a mouse model with a tamoxifen-inducible osteoblast-specific promoter (Col1a1) to induce the fluorescent protein Tomato (Tomato mice) expression. Again, co-expression of VE-cadherin and CD31 with Tomato was evident.
The authors verified the evidence and showed dexamethasone did not affect the CreER expression or Tomato activation. Dexamethasone did not activate the expression of endothelial markers in other tissues.
In OSX-Gfptg mice (another albeit redundant mouse model to the Tomato mice), the authors identified endothelial markers in the osteoblasts treated with dexamethasone
.
Isolated tomato-positive cells from DEX-treated tomato mice show decreased expression of osterix, osteopontin, and osteocalcin and induced expression of endothelial markers
. Transferring these cells to nude mice showed a reduced ectopic bone formation capacity compared to the tomato-positive cells from non-treated tomato mice. Using a hindlimb ischemia model, transplantation of Tomato-labelled osteoblasts demonstrated better perfusion when isolated from DEX-treated mice.

In vitro, analysis of the osteoblast cell line (MC3T3-E1) after treatment with various GCs demonstrated an increased EC-marker expression over time and decreased OB-marker expression.

Blocking Foxc2 in DEX-treated MC3T3-E1 abolished the induction of endothelial markers but failed to prevent the reduction of osteoblastic markers, whereas overexpression of Osterix prevented the decrease in osteoblastic markers but could not abolish endothelial markers.

Finally, the authors suggest that glucocorticoids suppress osteogenic capacity 391 and cause osteoporosis in part through this previously unrecognized transition of 392 osteoblast-lineage cells to endothelial differentiation.

This interesting and surprising study unravels an unwanted side effect of glucocorticoid treatment on osteoblast cells.

However, some questions emerged, and some minor points need to be addressed:

-          The title seems a bit overemphasizes the role of a possible cell transition in GC-induced bone loss. A GR-dim mouse model system or the transgenic expression of 11beta-hydroxysteroid dehydrogenase type 2 in osteoblasts should be applied to clarify the direct causal relationship of GC treatment on a possible cell transition.

-          COL1 is not only restrictively expressed in osteoblasts but also, albeit to a lesser extent, in other cells of mesenchymal origin, including mesenchymal stromal cells and smooth muscle cells.

-          Osteocalcin is a calcium-binding peptide hormone secreted by OBs but bound by various cells, including ECs. Is it possible that osteocalcin is attached to ECs expressing VE-cadherin and CD31? Please comment on that how you have controlled this possibility.

-          Moreover, GCs have been reported to reduce osteocalcin production. In that case, observing more cells being osteocalcin positive seems strange. Could you please comment?

-          To rule out that cell doublets are excluded, please provide a gating strategy.

-          The resolution of the IF figures is poor and does not rule out very close proximity after DEX treatment instead of transdifferentiation. Please provide high-resolution figures clearly distinguishing single cells.

-          In tomato mice, you found downregulation of osteocalcin while observing cells co-expressing either CD31 or VE-cadherin with osteocalcin. Please explain this conflicting observation. (see Fig1c, Fig2a/b versus Fig6b, Fig7b and Fig8a)

-          please revise the wording and English language carefully….

-          Methods for cell isolation for Ectopic bone formation and hindlimb ischemia are missing

-          Immunoblotting and IF are different methods requiring different antibodies. Should be separated under different headings

-          Information on centrifugation steps must be given in xg

-          To provide a translational character of the study, a simple approach using human cells should be given

please revise the wording and English language carefully….

Author Response

Reviewer 2

The manuscript entitled ”Cell Transitions Contribute to Glucocorticoid-Induced Bone Loss” by Qiao et al. addresses one major unwanted side effect of glucocorticoid treatment: glucocorticoid-induced osteoporosis, which may increase bone fracture risk. The authors describe that glucocorticoids convert osteoblast-lineage cells (OBs) into endothelial-like cells (ECs).The authors first identified the well-known osteoporotic effect of dexamethasone in C57BL/6J mice. They surprisingly identified endothelial markers VE-cadherin and CD31, as well as vWF, which emerged in the osteoblast lineage cells identified via osteocalcin and osterix. To identify the origin of the EC markers from osteoblasts, the authors used a mouse model with a tamoxifen-inducible osteoblast-specific promoter (Col1a1) to induce the fluorescent protein Tomato (Tomato mice) expression. Again, co-expression of VE-cadherin and CD31 with Tomato was evident. The authors verified the evidence and showed dexamethasone did not affect the CreER expression or Tomato activation. Dexamethasone did not activate the expression of endothelial markers in other tissues. In OSX-Gfptg mice (another albeit redundant mouse model to the Tomato mice), the authors identified endothelial markers in the osteoblasts treated with dexamethasone. Isolated tomato-positive cells from DEX-treated tomato mice show decreased expression of osterix, osteopontin, and osteocalcin and induced expression of endothelial markers. Transferring these cells to nude mice showed a reduced ectopic bone formation capacity compared to the tomato-positive cells from non-treated tomato mice. Using a hindlimb ischemia model, transplantation of Tomato-labelled osteoblasts demonstrated better perfusion when isolated from DEX-treated mice. In vitro, analysis of the osteoblast cell line (MC3T3-E1) after treatment with various GCs demonstrated an increased EC-marker expression over time and decreased OB-marker expression. Blocking Foxc2 in DEX-treated MC3T3-E1 abolished the induction of endothelial markers but failed to prevent the reduction of osteoblastic markers, whereas overexpression of Osterix prevented the decrease in osteoblastic markers but could not abolish endothelial markers. Finally, the authors suggest that glucocorticoids suppress osteogenic capacity 391 and cause osteoporosis in part through this previously unrecognized transition of 392 osteoblast-lineage cells to endothelial differentiation. This interesting and surprising study unravels an unwanted side effect of glucocorticoid treatment on osteoblast cells. However, some questions emerged, and some minor points need to be addressed:

  1. “The title seems a bit overemphasizes the role of a possible cell transition in GC-induced bone loss. A GR-dim mouse model system or the transgenic expression of 11beta-hydroxysteroid dehydrogenase type 2 in osteoblasts should be applied to clarify the direct causal relationship of GC treatment on a possible cell transition.”

Glucocorticoid-induced bone loss is directly caused by the intake of glucocorticoids. In this study, we treated mice or cells with glucocorticoids and found the transition of osteoblast-lineage cells. The results provided evidence supporting that glucocorticoid treatment is the causative factor for this cell transition. Either the GR-related model or the glucocorticoid inhibition model might partially support our study. However, using these models are out of scope of this study.

  1. “COL1 is not only restrictively expressed in osteoblasts but also, albeit to a lesser extent, in other cells of mesenchymal origin, including mesenchymal stromal cells and smooth muscle cells.”

Thanks for pointing this out. We have systemically determined the expression of Col1a1-driven CreER in multiple tissues and ruled out the involvement of marrow progenitors or mesenchymal cells from other organs. We stated “In addition to bone, Col1a1 is expressed in other tissues where CreER expression could be activated in the Col1a1cre/ERT2 mice [21]. In this study, we systemically examined the CreER expression in different organs, including bone, lungs, skin, heart, muscle, bone marrow, liver, kidneys, stomach, spleen, aorta, eyes, small intestine, pancreas, and brain of Col1a1cre/ERT2 mice, Col1a1cre/ERT2RosatdTomato mice, Col1a1cre/ERT2RosatdTomato mice with tamoxifen treatment and Col1a1cre/ERT2RosatdTomato mice with tamoxifen and dexamethasone treatment. Real-time PCR showed the highest expression of CreER in bone with only low levels of CreER expression in other tissues (Figure 4a), which confirmed a previous report that showed osteoblasts to have the highest CreER expression [21]. The results also revealed that the treatment with tamoxifen or dexamethasone had no effect on CreER expression in all these organs (Figure 4a).

To determine if tdTomato-positive marrow cells in bone or tdTomato-positive cells of other organs involved in the osteoblast-endothelial transition, we first used FACS to examine the percentage of tdTomato-positive cells in bone, lungs, skin, heart, muscle, bone marrow, liver, kidneys, stomach, spleen, aorta, eyes, small intestine, pancreas and brain of Col1a1cre/ERT2RosatdTomato mice, Col1a1cre/ERT2RosatdTomato mice with only tamoxifen treatment and Col1a1cre/ERT2RosatdTomato mice with tamoxifen and dexamethasone treatment. The results showed the highest percentage of tdTomato-positive cells in bone with only a low percentage of tdTomato-positive cells in other tissues of Col1a1cre/ERT2RosatdTomato mice with only tamoxifen treatment and Col1a1cre/ERT2RosatdTomato mice with tamoxifen and dexamethasone treatment (Figure 4b). The distribution pattern was similar to that of the CreER expression (Figure 4b), suggesting that tamoxifen efficiently moved CreER into cell nuclei for tdTomato activation. The results did not show a significant difference in the percent tdTomato-positive cells between dexamethasone treatment and control, suggesting that glucocorticoids had no effect on CreER translocation or tdTomato activation (Figure 4b). We then isolated tdTomato-positive cells from bone, lungs, skin, heart, muscle, bone marrow, liver, kidneys, stomach, spleen, and aorta, and examined the endothelial markers VE-cadherin, CD31, Flk1 and Foxc2. We only found induction of these markers in the tdTomato-positive cells from bone, not in cells from bone marrow or other tissues (Figure 4c). The results suggested that glucocorticoid-driven endothelial differentiation only occurred in tdTomato-labelled osteoblast-lineage cells and therefore ruled out the involvement of marrow progenitors or mesenchymal cells from other organs.”

  1. “Osteocalcin is a calcium-binding peptide hormone secreted by OBs but bound by various cells, including ECs. Is it possible that osteocalcin is attached to ECs expressing VE-cadherin and CD31? Please comment on that how you have controlled this possibility.”

We performed immunostaining and FACS using the same procedures for both the treated group and the controls. In the control group, there was no co-localization or co-expression between osteocalcin and the endothelial markers VE-cadherin, CD31 or vWF by immunostaining images or FACS (Figures 1 and 2). The results confirmed that osteocalcin was located in the normal osteoblasts, not ECs.

  1. “Moreover, GCs have been reported to reduce osteocalcin production. In that case, observing more cells being osteocalcin positive seems strange. Could you please comment?”

The FACS results showed a dramatically decreased cell population that only expressed osteocalcin after dexamethasone treatment (Figure 1 and 2), suggesting that, indeed, glucocorticoid reduced osteocalcin production in mature osteoblasts. We also observed a significant increase in the cell population that co-expressed osteocalcin and endothelial markers. Combined these two results suggested that glucocorticoids shift osteoblasts towards EC-like osteoblasts. Accumulation of osteocalcin-positive cells could result from recruitment of more progenitors that were feeding this cell transition. This hypothesis would be interesting for future studies but is out of the scope of this study.

  1. “To rule out that cell doublets are excluded, please provide a gating strategy.”

The gating strategy was added to the FACS section of the Methods.

  1. “The resolution of the IF figures is poor and does not rule out very close proximity after DEX treatment instead of transdifferentiation. Please provide high-resolution figures clearly distinguishing single cells.”

We added DAPI staining to visualize the nuclei (Figure 2b). In the image, individual osteoblasts and ECs could be identified in the controls. However, glucocorticoid treatment significantly altered the normal bone structure, cell shape and gene expression. It is extremely difficult to find structures that are maintained similarly to the controls.

In addition, we added time-course FACS analysis of demarrowed femurs from dexamethasone-treated mice using anti-CD31, anti-VE-cadherin, anti-Flk1 or anti-Osteocalcin antibodies (Figure 2c). We performed real-time PCR and showed decreased expression of osteoblast lineage markers (Figure 2e), which were shown to be down-regulated in osteocalcin-positive cells after dexamethasone-treatment (Figure 2d).

  1. “In tomato mice, you found downregulation of osteocalcin while observing cells co-expressing either CD31 or VE-cadherin with osteocalcin. Please explain this conflicting observation. (see Fig1c, Fig2a/b versus Fig6b, Fig7b and Fig8a)”

In Figures 1 and 2, we detected certain levels of osteocalcin protein, which might have been expressed before or during glucocorticoid treatment. The results showed that osteocalcin protein could remain for certain time periods. However, in Figure 6 and Figure 7, the data was obtained using osteocalcin transcripts, which were affected shortly after the glucocorticoid treatment. In Figure 8, it was a cell line that was treated in vitro. Interestingly, the results suggested that retention of osteocalcin in cells might be different between in vivo and in vivo in responses to glucocorticoid treatment. Also, it could be the result of different detection methods as immunostaining and FACS are more sensitive than immunoblotting.

  1. “please revise the wording and English language carefully….”

We have edited the manuscript. 

  1. “Methods for cell isolation for Ectopic bone formation and hindlimb ischemia are missing”.

Methods for cell isolation were included in the FACS section of the Methods.

  1. “Immunoblotting and IF are different methods requiring different antibodies. Should be separated under different headings”

It was updated.

  1. “Information on centrifugation steps must be given in xg”

It was updated.

  1. “To provide a translational character of the study, a simple approach using human cells should be given.”

We obtained a human osteoblast line (hFOB1.19) and treated the cells with 10-7 M dexamethasone. The real-time PCR showed similar pattern of time-course expression of endothelial and osteogenic markers after dexamethasone treatment (Figure 7c).

Reviewer 3 Report

In this study, the authors investigated the process of osteoblastic-endothelial transition in glucocorticoid-induced bone loss using two types of mice from the osteoblastic lineage: Col1a1cre/ERT2 Rosa tdTomato mice and OSX-Gfptg mice. Additionally, they identified two key regulators, Foxc2 and Osterix, that drive this transition. The authors present a new and comprehensive perspective on glucocorticoid-induced bone Loss. However, there are a few concerns that should be addressed before considering it for publication.

1.Figure 2 only includes a high-magnification image of anti-OSX. It would be beneficial to provide more data in this figure.

 2.In Figure 6a, the population of tdTomato-positive cells appears to be similar between the control and dexamethasone treatment groups. However, the expression of Col1a1 is decreased with dexamethasone treatment. It is important to explain this data discrepancy in order to provide a better understanding of the results.

 3.It would be valuable to include additional assays, such as histology and immunohistochemistry, to confirm the claim that "Glucocorticoid-Treated Osteoblasts Lose Osteogenic Capacity in Ectopic Bone Formation but Improve Vascular Repair" (Figure 6).

4.The discussion section should be reorganized. By integrating the findings with existing literature, the discussion section would become more coherent and provide a clearer analysis of the research.

Author Response

Reviewer 3

In this study, the authors investigated the process of osteoblastic-endothelial transition in glucocorticoid-induced bone loss using two types of mice from the osteoblastic lineage: Col1a1cre/ERT2 Rosa tdTomato mice and OSX-Gfptg mice. Additionally, they identified two key regulators, Foxc2 and Osterix, that drive this transition. The authors present a new and comprehensive perspective on glucocorticoid-induced bone Loss. However, there are a few concerns that should be addressed before considering it for publication.

  1. “Figure 2 only includes a high-magnification image of anti-OSX. It would be beneficial to provide more data in this figure.” 

In revised Figure 2, we added time-course FACS analysis of demarrowed femurs of dexamethasone-treated mice using anti-CD31, anti-VE-cadherin, anti-Flk1 or anti-Osteocalcin antibodies (Figure 2c). We performed real-time PCR and showed the decreased expression of osteoblast lineage markers (Figure 2e), which were identified to be down-regulated in osteocalcin-positive cells after dexamethasone-treatment (Figure 2d). We also showed images of the co-expression of Osteocalcin with CD31, VE-cadherin and vWF in the femurs of dexamethasone-treated mice (Figure 2b).

  1. “In Figure 6a, the population of tdTomato-positive cells appears to be similar between the control and dexamethasone treatment groups. However, the expression of Col1a1 is decreased with dexamethasone treatment. It is important to explain this data discrepancy in order to provide a better understanding of the results.”

We added the following to the text. We added “Since the tdTomato expression was induced prior to the dexamethasone treatments, the results showed the similar tdTomato levels between treated group and non-treated control, suggesting similar efficiency of tdTomato labelling in osteoblast-lineage cells in these groups.”

  1. “It would be valuable to include additional assays, such as histology and immunohistochemistry, to confirm the claim that "Glucocorticoid-Treated Osteoblasts Lose Osteogenic Capacity in Ectopic Bone Formation but Improve Vascular Repair" (Figure 6).”

       We performed immunostaining using anti-CD31 antibodies to quantitate CD31+ cells in the implants of tdTomato-labelled osteoblasts. The results showed more CD31+ cells in the implants of the dexamethasone-treated group that the non-treated control (Figure 6d). We also examined the vascularization at the sites of transplantation of tdTomato-labelled cells using latex dye with tissue clearing, which showed stronger vascular growth in the dexamethasone-treated group than that of the non-treated control (Figure 6g).

  1. “The discussion section should be reorganized. By integrating the findings with existing literature, the discussion section would become more coherent and provide a clearer analysis of the research.”

        We have revised the discussion.

Round 2

Reviewer 1 Report

The authors have addressed all of my concerns.